# Zoledronic Acid as a Novel Dual Blocker of KIR6.1/2-SUR2 Subunits of ATP-Sensitive K^+^ Channels: Role in the Adverse Drug Reactions

**DOI:** 10.3390/pharmaceutics13091350

**Published:** 2021-08-27

**Authors:** Fatima Maqoud, Rosa Scala, Vincenzo Tragni, Ciro Leonardo Pierri, Maria Grazia Perrone, Antonio Scilimati, Domenico Tricarico

**Affiliations:** 1Section of Pharmacology, Department of Pharmacy-Pharmaceutical Sciences, University of Bari, Via Orabona 4, 70125 Bari, Italy; Fatima.maqoud@uniba.it (F.M.); Rosa.scala@uniba.it (R.S.); 2Laboratory of Biochemistry, Structural and Molecular Biology, Department of Biosciences, Biotechnologies and Biopharmaceutics, University of Bari, Via Orabona 4, 70125 Bari, Italy; vincenzo.tragni@uniba.it (V.T.); ciro.pierri@uniba.it (C.L.P.); 3BROWSer S.r.l., University of Bari “Aldo Moro”, Via E. Orabona, 4, 70126 Bari, Italy; 4Medicinal Chemistry Section, Department of Pharmacy-Pharmaceutical Sciences, University of Bari, Via Orabona 4, 70125 Bari, Italy; mariagrazia.perrone@uniba.it (M.G.P.); antonio.scilimati@uniba.it (A.S.)

**Keywords:** zoledronic acid, adverse drug reactions, musculoskeletal and cardiovascular, KATP channels

## Abstract

Zoledronic acid (ZOL) is used as a bone-specific antiresorptive drug with antimyeloma effects. Adverse drug reactions (A.D.R.) are associated with ZOL-therapy, whose mechanics are unknown. ZOL is a nitrogen-containing molecule whose structure shows similarities with nucleotides, ligands of ATP-sensitive K^+^ (KATP) channels. We investigated the action of ZOL by performing in vitro patch-clamp experiments on native KATP channels in murine skeletal muscle fibers, bone cells, and recombinant subunits in cell lines, and by in silico docking the nucleotide site on KIR and SUR, as well as the glibenclamide site. ZOL fully inhibited the KATP currents recorded in excised macro-patches from Extensor digitorum longus (EDL) and Soleus (SOL) muscle fibers with an IC_50_ of 1.2 ± 1.4 × 10^−6^ and 2.1 ± 3.7 × 10^−10^ M, respectively, and the KATP currents recorded in cell-attached patches from primary long bone cells with an IC_50_ of 1.6 ± 2.8 × 10^−10^ M. ZOL fully inhibited a whole-cell KATP channel current of recombinant KIR6.1-SUR2B and KIR6.2-SUR2A subunits expressed in HEK293 cells with an IC_50_ of 3.9 ± 2.7 × 10^−10^ M and 7.1 ± 3.1 × 10^−6^ M, respectively. The rank order of potency in inhibiting the KATP currents was: KIR6.1-SUR2B/SOL-KATP/osteoblast-KATP > KIR6.2-SUR2A/EDL-KATP >>> KIR6.2-SUR1 and KIR6.1-SUR1. Docking investigation revealed that the drug binds to the ADP/ATP sites on KIR6.1/2 and SUR2A/B and on the sulfonylureas site showing low binding energy <6 Kcal/mol for the KIR6.1/2-SUR2 subunits vs. the <4 Kcal/mol for the KIR6.2-SUR1. The IC_50_ of ZOL to inhibit the KIR6.1/2-SUR2A/B channels were correlated with its musculoskeletal and cardiovascular risks. We first showed that ZOL blocks at subnanomolar concentration musculoskeletal KATP channels and cardiac and vascular KIR6.2/1-SUR2 channels.

## 1. Introduction

Bisphopshonates (BPs) are the most used antiresorptive drugs, useful for the treatment of bone diseases associated with an increased resorption rate. They include compounds without nitrogen atoms in their structure, such as etidronate and clodronate that are expected to cause osteoclast apoptosis via the formation of toxic ATP-metabolites [1], and nitrogen-containing BPs, such as pamidronic acid, alendronic acid, risedronic acid, and zoledronic acid. Once internalized into the osteoclasts, nitrogen-containing BPs can act as potent inhibitors of human farnesyl pyrophosphate synthase (*h*FPPS), a key enzyme in the mevalonate pathway, inducing osteoclast apoptosis [2,3,4]. In our previous experiments, zoledronic acid (ZOL) was found to be the most potent compound in inhibiting the activity of the purified recombinant *h*FPPS, showing an IC_50_ of 0.06 ± 0.02 × 10^−3^ M, and in binding the hydroxyapatite [5,6], also displaying antiproliferative effects against pre-osteoclast like cells RAW264.7 and J774A.1 and tumor cells MG63 and PC3, as well as against fully differentiated osteoclasts [6].

BPs are also known for their antimyeloma and antitumor effects. They can inhibit the osteolysis coupled with tumor growth and the growth, migration, and matrix-associated invasion of breast cancer cells, leading to antimetastatic effects [7]. In multiple myeloma patients, BPs reduce pathological vertebral fractures, skeletal-related events (SREs), and pain. ZOL was found more effective than placebo and first-generation etidronate for improving overall survival and other outcomes, such as vertebral fractures, [8] and superior to clodronic acid for both progression-free and overall survival, with a median overall survival of 52 months [9].

However, dark aspects of BPs remain. Several adverse drug reactions (A.D.R.) of ZOL and other BPs are reported in the literature [10,11,12] and in the EPAR related documents (https://www.ema.europa.eu/en/medicines/human/EPAR/zometa (accessed on 9 July 2021)). Among these, gastric irritation, osteonecrosis of the jaw, atypical femoral fractures, esophageal cancer, atrial fibrillation, severe musculoskeletal pain and weakness, and atypical femur fractures are observed in the disease patients population under treatment with ZOL [13]. Zoledronic acid was one of the most common medications inducing arrhythmias in the Food and Drug Administration Adverse Event Reporting System (F.A.E.R.S.) database [14]. Interestingly, many of these A.D.R. is nowadays unexplained in their molecular mechanisms, thus suggesting the involvement of unidentified pathways [12]. In particular, BPs are known to highly accumulate in bone. Here, and in further soft tissues like the skeletal muscle surrounding the bone, the direct interaction between BPs and additional targets responsible for side effects cannot be excluded.

We have recently shown that ZOL at micromolar concentrations can directly activate TRPV1 ion channels in osteoblasts and neurons, but not in osteoclasts, and that the ZOL-mediated activation of this channel is required for pro-survival/pro-mineralizing effects in osteoblasts and possibly for pain modulation [15].

Among these, ATP-sensitive potassium (KATP) channels are known to be expressed in several tissues, even included the ones involved in BPs therapeutical and adverse effects. The available KATP channels structures (5wua.pdb [16]; 5ywb.pdb; and 5ykf.pdb, [17] show that four SUR regulatory subunits locate peripherally and dock onto the central Kir channel tetramer [16,17]. Each SUR and/or Kir subunit in the available cryo-em structures shows an adenine nucleotide derivative in a well-defined binding region (5ywb.pdb; and 5ykf.pdb, [17]. Furthermore, SUR subunit structure in complex with glibenclamide is also available on the protein data bank (5ykf.pdb, [17]. The composition of the Kir/SUR subunits forming the KATP channels depends on the different tissue expression pattern of the Kir/SUR subunits.

In skeletal muscle and cardiomyocytes, KATP channels are mostly composed of KIR6.2-SUR2A subunits. In endothelial vascular smooth muscle, KATP channels are composed of KIR6.1-SUR2B subunits. The KIR6.2-SUR1 compose the main channel complex in pancreatic beta cells, whereas, in the bone, the composition is uncertain. The function and composition of the channel subunits, however, is more complex than expected, and different subunit composition and distribution have been found in various tissues. KATP channels have a well-known physiopathological role in regulating vascular tone, cardiac excitability, and neuronal firing. The insulin release from the pancreas, muscle fatigue and pain, and neurodegeneration [18,19,20,21,22,23,24,25] are also involved in non-familial diseases and in rare KATP channelopathies [21,26]

KATP channels are regulated by intracellular nucleotides such as ATP and ADP that finely tune the KATP channel’s activity in response to metabolic stress. Interestingly, ZOL is a nitrogen-containing bis-phonate ligand consisting of two phosphate groups and one imidazole group that may overlap with the phosphate group and the purine/ribose ring of these nucleotides.

In this work, we tested the effects of ZOL on KATP channels by performing patch-clamp experiments in native murine fast-twitching and slow-twitching muscle fibers, primary bone cells culture, and recombinant channel subunits expressed in heterologous cells system. Furthermore, we performed docking studies to evaluate the in silico interaction of ZOL with KATP channel subunits and to identify the possible binding site. Finally, a pharmacovigilance investigation was performed to evaluate the musculoskeletal A.D.R. of ZOL associated with KATP channel interaction.

### Significant Statement

Zoledronic acid is extensively prescribed in therapy. Because of the numerous benefits associated with its use, zoledronic acid has been included in the essential medicine list of W.H.O. 2019 for the cure of malignancy-related bone disease. Its main pharmacological properties are related to antiproliferative action against malignant cells/osteoclasts and proliferative actions on osteoblasts. Adverse effects associated with the extensive use of this drug have been reported by the patients, remaining unexplained. We correlated the blocking action of zoledronic acid toward KIR6.1/2-SUR2A/B subunits of the KATP channel with invalidating musculoskeletal and cardiovascular reactions. The knowledge about this action is worthy of interest to ameliorate prescription in preventing arrhythmias, weakness, rhabdomyolysis, and pain. For instance, by using KATP openers.

## 2. Materials and Methods

### 2.1. Tissues and Primary Cell Culture

In accordance with 3R RULE (DIRECTIVE 2010/63/EU) Replace, Reduce, Refine, bone and skeletal muscle were collected from wild-type male mice C57BL sacrificed for other scientific purposes (Ethic Committee name, Organization for Animal Health O.P.B.A. University of Bari, Italy, project: 7282018PR). The fibers were enzymatically isolated from Extensor digitorum longus (EDL) and Soleus (SOL) muscles, as previously described [27,28].

The culture of primary bone cells from adult mice was obtained from long bones (tibia and femur) and calvaria, as previously described [29]. Bones were collected, cleaned, and flushed for removing the internal bone marrow cells. Small bone pieces of around 1 mm^3^ were treated with trypsin-EDTA 0.25% (*w*/*v*) for 1 h and with 0.2% collagenase solution for an additional hour in a shaking water bath to remove all remaining soft tissue and adherent cells. Clean bone pieces have been cultured in basal medium enriched with 50 µg/mL of ascorbic acid. Osteoblasts from femora (Appendix A) and calvaria chips (Appendix A) were positive to Alizarin red staining (Appendix A). The marker of osteoblasts was positive, as tested by the PCR expression analysis on cell pellets (Appendix A).

For experiments on recombinant channel subunits, Hek293 cells were transfected with the relevant plasmids, as previously described [30].

### 2.2. Drugs and Solutions

KATP channels modulators glyburide/glibenclamide cat. N° PHR1287, repaglinide cat. N° R9028, tolbutamide cat. N° T0891, and BaCl_2_ cat. N° 449644 were purchased from Sigma (SIGMA Chemical Co., Milan, Italy). The zoledronic acid (ZOL) was synthesized and purified in our labs, as previously described [6], and dissolved in phosphate-buffered saline (PBS) stock solution at 1 or 10 × 10^−3^ M.

The normal Ringer solution used during muscles and organ biopsy contained 145 mM NaCl, 5 mM KCl, 1 mM MgCl_2_, 0.5 mM CaCl_2_, 5 mM glucose, and 10 mM 3-(N-morpholino) propanesulfonate (Mops) sodium salt and was adjusted to pH 7.2 with Mops acid.

The solutions for excised patch experiments on isolated muscle fibers were as follows. The patch-pipette solution contained 150 mM KCl, 2 mM CaCl_2_, and 1 mM Mops (pH 7.2); the bath solution contained 150 mM KCl, 5 mM EGTA, and 10 mM Mops (pH 7.2).

The solutions for whole-cell and cell attached patches were as follows. The pipette solution contained 132 mM KCl, 1 mM ethylene glycol-bis(β-aminoethylether)-*N*,*N*,*N′*,*N*′-tetraaceticacid (EGTA), 10 mM NaCl, 2 mM MgCl_2_, 10 mM HEPES, 1 mM Na_2_ATP, and 0.3 mM Na_2_GDP (pH = 7.2). The bath solution contained 142 mM NaCl, 2.8 mM KCl, 1 mM CaCl_2_, 1 mM MgCl_2_, 11 mM glucose, and 10 mM HEPES (pH = 7.4).

CaCl_2_ was added to the pipette solutions to give a free Ca^2+^ ion concentration of 1.6 × 10^−6^ M in whole-cell experiments. The calculation of the free Ca^2+^ ion concentration in the pipette was performed using the MaxChelator software (Stanford University; Stanford, CA, USA).

### 2.3. Patch-Clamp Experiments

The whole-cell and cell-attached patch-clamp experiments were performed in asymmetrical K^+^ ion concentration in physiological conditions using pipettes with the resistance of 3–5 MΩ.

The KATP currents of native skeletal muscle fibers were recorded in excised macro patches using pipettes of 0.9–1.2 MΩ of resistance on isolated skeletal muscle fibers in symmetrical 150 mM K^+^ ions concentrations on both sides of the membrane during pulse going from 0 mV to −60 mV (Vm), as previously described [31,32].

Drug actions on the channel currents recorded during instantaneous I/V relationships were investigated by applying a depolarization protocol, in response to voltage pulses from −180 mV to +100 mV (Vm) in 20 mV steps. Currents were expressed as densities (pA/pF) to control for cell size/capacitance differences. All experiments were performed at room temperature (20–22 °C) and sampled at 2 kHz (filter = 1 kHz) using an Axopatch-1D amplifier equipped with a CV-4 headstage (Axon Instruments; Foster City, CA, USA). Drug solutions were applied using a fast perfusion system during the continuous monitoring of the seal resistance at 18–22 °C room temperature in excised patches. In whole-cell and cell-attached patch-clamp experiments, before recording, the cells were equilibrated for 10 s with the solution. Data acquisition and analysis were performed using pCLAMP 10 software suite (Axon Instruments; Foster City, CA, USA), as previously described. Seal resistance was continuously monitored during the experiment.

### 2.4. Polymerase Chain Reaction

Total RNA was isolated and purified from the entire EDL and SOL muscles and the calvaria and femora bone with Trizol reagent (Invitrogen, Thermo Fisher Scientific Inc., Waltham, MA, USA) and quantified using a spectrophotometer (ND-1000 Nano-Drop, Thermo Fisher Scientific Inc., Waltham, MA, USA). PCR amplification was achieved using PCR Master Mix (Promega Italia S.r.l., Milano, Italy). PCR cycles consisted of denaturation at 95 °C for 1 min, annealing segment at 58 °C for 1 min, and extension at 72 °C for 1 min, repeated for 30 cycles. Amplified PCR products were separated on 1% agarose gel. The data collection and analysis were performed according to the MIQE. All experiments were performed in triplicate per tissues.

Primary sequences 5′–3′: *Abcc8* F ATCATTCTGCTGGCTCCTGT, R CTGGTCATTTCCTTCCTGCG; *Abcc9* F CTGGTCCCACATGTCTTCCT, R ATGCGAGTCTGAAACGATGC; *Kcnj8* F GTAGACCTGAAGTGGCGTCA, R GCATGGCGGCTGAAAATCA; *Kcnj11* F CACCTCCTACCTAGCTGACG, R ATGCTAAACTTGGGCTTGGC;*Alp* F GTTGCCAAGCTGGGAAGAACAC, R CCCACCCCGCTATTCCAAAC; *Bglap* F CGCTACCTGTATCAAT, R CTCCTGAAAGCCGATG; *Msx2*, F TGAGGAAACACAAGACCAA, R 5-GTCTATGGAAGGGGTAGGAT.

### 2.5. Molecular Modeling Analysis

A 3D comparative model for the human KIR6.1 (NP_004973.1), KIR6.2 (NP_000516.3), SUR1 (NP_000343.2), SUR2A (NP_005682.2), and SUR2B (NP_001364202.1) proteins was built using [33] Modeller [REF. MODELLER].

The 3D crystallized structures of KIR6.2 (from *M. musculus*) and SUR1 (from *M. auratus*) reported in the protein complexes 5wua.pdb, 5ywb.pdb, and 5ykf.pdb were used as protein templates.

A sequence structure pairwise alignment between the sequence of the human proteins to be modeled and the most similar crystallized structure selected among the above-cited PDB entries was obtained using ClustalW [34] and used for driving the comparative modeling session [35,36].

The obtained 3D Kir and SUR comparative models were superimposed to the crystallized counterpart from 5wua.pdb (KIR6.2-SUR1 complex), 5ywb.pdb (KIR6.2-SUR1, complexed with ADP and Mg^2+^), and 5ykf.pdb (KIR6.2-SUR1, complexed with 5-chloro-N-(2-{4-[(cyclohexylcarbamoyl)sulfamoyl]phenyl}ethyl)-2-methoxybenzamide, also known as glibenclamide (Glib), and phosphothiophosphoric acid adenylate ester (AGS), the latter mimicking an ATP molecule) [16,17] protein-ligand complexes for obtaining the human heterodimeric protein complexes KIR6.1-SUR2B, KIR6.2-SUR1, KIR6.2-SUR2A, and KIR6.1-SUR1 using the super command available in the PyMOL suite according to validated protocols [37,38].

Ligands ADP and Glib were duplicated from the above-cited protein-ligand crystallized structures and docked into the human KIR6.1-SUR2B, KIR6.2-SUR1, KIR6.2-SUR2A and KIR6.1-SUR1 protein complexes in the corresponding binding regions after the superimposition steps, as previously described [37,38]).

All the generated 3D all-atom models were energetically minimized using the Rosetta “relax” script from the Rosetta “scoring and prep” tools [39,40] (see also the tutorial: https://www.rosettacommons.org/demos/latest/tutorials/scoring_and_prep/scoring_and_prep, accessed on 12 December 2020), according to the protocols previously described [41]. The structural properties of the generated 3D atom models with the best energy function were evaluated using the Whatif webserver to check the ψ, ϕ, and Cβ angles of each residue [42]. The obtained final models were examined in VMD [43], PyMOL [44], and SPDBV [45] by visual inspection searching for putative unsolved clashes, according to [35,36,37,38,41].

Zoledronic acid, ATP isopentenyl ester, and ATP were docked within the ATP binding region at the KIR subunit and the SUR subunit using Autodock4.2 [46]. In addition, zoledronic acid was also docked within the Sur-Glib binding region. With this aim, three grid boxes to be explored through docking runs were prepared and consisted of all residues within 4 Å from ADP (in SUR1, SUR2A, or SUR2B; grid box 1), ADP (in KIR6.2 or KIR6.1, grid box 2), and Glib (in SUR1, SUR2A, or SUR2B, grid box 3), respectively. The center of each grid box was put in the center of the mass of ADP (x = 129.506; y = 87.669; and z = 136.068) or Glib (x = 129.86; y = 108.775; and z = 180.245) for the grid box within the human Sur corresponding subunits, or ADP (x = 133.475; y = 166.938; and z = 163.919) for the grid box within the human KIR subunits. The number of grid points for grid boxes 1, 2, and 3 along the axes x, y, and z was 38x.52y.72z; 50x.68y.72z; and 42x.50y.40z, respectively.

Before running the docking analysis, a re-docking simulation was performed for validating the docking strategy. The root-mean-square-deviation (RMSD) between the re-docked ADP and the ADP ligand present in the crystallized SUR1 (ADP, 5ywb.pdb), the re-docked Glib and the Glib ligand present in the crystallized SUR1 (5ykf.pdb), and the re-docked ADP and the ADP ligand present in KIR6.2 or SUR1 (5ywb.pdb) was 1.48 Å, 0.98 Å, 1.85 Å, respectively.

### 2.6. Pharmacovigilance Analysis

All coding preferred terms per System Organ Class (S.O.C.) were screened in the EudraVigilance database (http://www.adrreports.eu/ accessed on 1 May 2021), and those potentially associated with KATP channel blocking actions were reported and subject to analysis.

The data were extracted from EudraVigilance, collected, and analyzed on Microsoft 365 Excel software package (Microsoft 10.00). The not specified category was not included in the calculation. The duplication of a specific report per S.O.C. was evaluated manually and excluded from the analysis [32].

### 2.7. Statistical Analysis

The data are expressed as an average ± E.S. unless otherwise specified. The significance between data pairs was calculated by the paired *t-student* test for *p* < 0.05. One Way ANOVA was used to evaluate the significance within and between data with a variance-ratio F > 1 at significant levels of *p* < 0.05.

The percentage of KATP current inhibition induced by ZOL was calculated as (I CTRL-I drug)/(I CTRL-I BaCl_2_) × −100. The concentration response data of ZOL against different KATP channel currents were fitted by an algorithm using the non-linear dynamic fitting routine of Sigma Plot, 10.0.

The Proportional Reporting Ratio (P.R.R.) was calculated using the following equation: a/(a+c)/b/(b + d) 
where a is the reaction of interest to a given drug of interest, b is the reaction of interest for all other drugs in the class, c are all other reactions to a given drug of interest, and d are all other reactions to all other drugs in the class. Signal definition: P.R.R. ≥ 2, a minimum of three ratios/cases for the reaction of interest, X^2^ ≥ 4. No signal is identified if P.R.R. is = 1 [47].

The linear correlation analysis and multiple correlation analysis between variables were performed using: y = aln(x) + b and multiple correlation equation: y = b0 + blnx1 + b2x^2^, respectively, as previously described [19].

The correlation analysis was performed between the mean values of the IC_50_ data of different KATP channel blockers and their relative P.R.R. The IC_50_ data of different KATP channel blockers were pooled from the literature and obtained from concentration-response curves of transmembrane KATP channel currents vs. at least five to seven different drug concentrations. The IC_50_ values were derived from the standard formula and reflect the drug concentration that would inhibit 50% of KATP channel current when measured in a drug-free solution and expressed as fractional current. The data were obtained in a variety of cell lines and sources for SUR and KATP channel proteins to obtain the IC_50_ values from different labs, as previously reported [48]. PCR experiments were performed in triplicate per tissue and presented as a sample PCR gel. Patch-clamp experiments were performed in >13 cells.

## 3. Results

### 3.1. Zoledronic Acid Inhibits KATP Channel Currents in Native Murine Skeletal Muscle, Bone Cells, and Recombinant Subunits Expressed in Cell Lines

Preliminary unpublished observations in our labs revealed that ZOL can reduce inward currents in whole-cells patch-clamp experiments performed in physiological conditions at sub-nanomolar concentrations, thereby suggesting that ZOL can also modulate different ion channels.

In patch-clamp experiments performed on excised macro-patches from fast-twitch Extensor digitorum longus (EDL) and slow-twitch Soleus (SOL) murine muscle fibers, we found that increasing concentrations of ZOL (10^−10^–10^−7^ M) solutions applied on the internal side of the membrane inhibited the KATP currents (Figure 1A,B). ZOL was more effective in inhibiting the KATP currents in SOL vs. EDL fibers.

In skeletal muscle fibers, KATP currents were also inhibited by glibenclamide (Glib) that markedly reduced the KATP currents in C-A patches at nanomolar concentrations (Figure 1C). A similar inhibitory response was observed with repaglinide (*n* = 2 cells) but not with tolbutamide, which was a less effective inhibitor of these currents (*n* = 3 cells).

Moreover, ZOL (10^−10^–10^−7^ M) was able to inhibit the currents recorded in the cell-attached patches of primary bone cells from long bones or calvaria, either at positive or negative membrane potential (Figure 1C). ZOL caused a full reduction of the channel current at subnanomolar concentrations, similar to SOL fibers. PCR analysis revealed an abundant presence of *Abcc9* and *KcnJ11* in EDL but a lower expression level in SOL muscles, as previously reported [31] (Figure 1D). Furthermore, we found an abundant presence of *Kcnj8* in either bone cells or bone tissue (Figure 1D).

Different potencies have been observed after the application of ZOL (10^−12^–10^−4^ M) to EDL, SOL, and primary bone cells from the femora (Figure 2A). The IC_50_ values of ZOL in the primary bone cells from femora were comparable with that of this drug in SOL fibers (Table 1). Moreover, in these cells, the application of glibenclamide (10^−12^–10^−4^ M) concentration-dependently inhibited the KATP currents in C-A patches. The IC_50_ of glibenclamide to inhibit the KATP current in primary bone cells was comparable with that of SOL fibers (Table 1). The potency and efficacy of Glib in different skeletal muscles phenotypes have already been extensively investigated, and the IC_50_ values have already been shown to be in the nanomolar concentration range [28,31,47].

ZOL (10^−12^–10^−3^ M) inhibited the whole-cell inward currents recorded in asymmetrical K^+^ ions condition of recombinant KATP channels subunits expressed in Hek293 cells (Figure 2B). The rank order of potency based on the calculated IC_50_ values and maximal efficacy on inward currents measured at −60 mV (Vm) was: KIR6.1-SUR2B > KIR6.2-SUR2A > KIR6.2-SUR1 and KIR6.1-SUR1 (Table 1). Therefore, high-affinity binding sites for ZOL are expected to be located on the KIR6.1/2-SUR2B/A subunits, while low-affinity binding sites could be located on KIR6.2-SUR1 subunits.

#### 3.1.1. Docking Analysis

The percentage of identical residues between the human SUR1, SUR2A, or SUR2B and the crystallized *M. auratus* Sur1 was 95% (SUR1) and 68% (for both SUR2A and 2B), respectively, according to ClustalW estimations [35,36]. The percentage of identical residues between the human KIR6.2 or KIR6.1 and the crystallized *M. musculus* KIR6.2 was 96% and 67%, respectively.

The RMSD between the atomic coordinates of the 3D relaxed models of the human Sur2A, Sur2B, or Sur1 and the crystallized *M. auratus* Sur1 domain was 0.28 Å (5wua.pdb, chain G), 1.81 Å (5ykf, chain F) or 3.7 Å (5ywb.pdb, chain F), or 2.82 Å (5wua.pdb, chain G), 3.64 Å (5ykf, chain F) or 3.95 Å (5ywb.pdb, chain F), respectively, whereas the RMSD between the atomic coordinates of the 3D relaxed models of the human KIR6.1 or KIR6.2 and the crystallized *M. musculus* KIR6.2 was 0.26 Å (5wua.pdb, chain D), 0.86 Å (5ykf, chain E) or 1.76 Å (5ywb.pdb, chain E), or 0.26 Å (5wua.pdb, chain G), 0.89 Å (5ykf, chain E) or 1.81 Å (5ywb.pdb, chain E), respectively.

Residues within 4 Å from the docked ligands in the human SUR1 or SUR2A or the human KIR6.2 or KIR6.1 are reported in Figure 3A,B and Appendix A.

Our docking simulations reveal a possible binding of ZOL to the ATP binding region, to the SUR subunit ADP binding region, and to the SUR subunit Glib binding region in all complexes (Table 2).

#### 3.1.2. Pharmacovigilance Analysis

We finally analyzed the profile of A.D.R. associated with ZOL therapy. The adverse reaction data of ZOL (all indications) showed that it is responsible for 35,201 cases reported in the EudraVigilance database up to 1 May 2021; 67.2% of the affected people were female, of whom 37.6% were adult (18–65 years old), 52.9% were a part of the adult-aged population (65–85 years old people), and 9% were overaged people >85 year old (0.5% not specified).

The most frequently observed A.D.R. reported per System Organ Class (S.O.C.) were musculoskeletal and connective tissues disorders, general disorder and administration site condition, and gastrointestinal disorders. Among the musculoskeletal and connective tissues disorders, the osteonecrosis of the jaw was the most frequently reported A.D.R., representing 47.9% of the total A.D.R. and affecting 4792 females and 3184 males of all ages. Arthralgia and myalgia were also frequently reported representing 16% and 10% of the A.D.R., respectively, with several unresolved cases. Muscular weakness was 2.7% of the A.D.R. per S.O.C., and some cases of atrophy and rhabdomyolysis with blood creatine phosphokinase increased were reported.

Within the vascular S.O.C., the most frequently observed A.D.R. was hypertension (25%) and blood pressure increase (16%) with systolic impairment, as compared to hypotension (18%), which was less frequently reported. Cases of ischemia, peripheral artery occlusion, and peripheral vascular disorders have been also observed.

Atrial fibrillation was frequently observed among the cardiac A.D.R. (17.9%), followed by palpitations (13.2%), tachycardia (12.2%), sinus tachycardia (5.3%), and arrhythmia 197 (8.1%) with heart rate increased, and sometimes associated with QT prolongation. Myocardial infarction (12%) and congestive heart failure (12.6%) were not uncommon within the cardiac A.D.R. affecting the general adult-aged patient population with cases of cardiac arrest in the oncologic and non-oncologic indications.

Hypoglycemia represented only 1.63% of the A.D.R. per S.O.C. of ZOL associated with diabetes mellitus and glucose intolerance. The Proportional Reporting Ratio (P.R.R.) values indicating the relative risk of ZOL of inducing A.D.R. within the KATP channel blockers showed that ZOL had P.R.R. values ≥2 for osteonecrosis of jaw > muscular weakness > myalgia > hypertension > arrhythmias > diarrhea > atrial fibrillation, and lower risk for rhabdomyolysis and hypoglycemia, despite several reported cases (Figure 4).

As expected, hypoglycemia was instead the most frequently observed A.D.R. of sulfonylureas and glinides, representing, for instance, the 84.6%, 83.13%, 77.9%, and 75.6% of A.D.R. for repaglinide, glibenclamide, nateglinide, and glimepiride, respectively. Glibenclamide also showed some hypoglycemia cases in neonatal, pediatric, and adolescent patients affected by neonatal diabetes and other diabetic disorders. Drugs showing the highest P.R.R. values ≥2 for hypoglycemia were: glibenclamide > glimepiride ≥ repaglinide > nateglinide.

Within the musculoskeletal and connective tissue disorders, rhabdomyolysis was reported in the 21.3%, 18.4%, 17.8%, and 17.7% for glibenclamide, nateglinide, repaglinide, and glimepiride, respectively. The highest risk for rhabdomyolysis within the KATP channel blockers was calculated for nateglinide >> glimepiride > glibenclamide. Lower P.R.R. values were calculated for the other drugs including ZOL. Myalgia and muscular weakness were also frequently observed, while muscle atrophy was less commonly reported. The P.R.R. values of muscular weakness were ≥2 for gliclazide > glibenclamide. Rare cases of osteonecrosis of the jaw and osteoporosis were also reported with glibenclamide and glimepiride.

Despite several cases reported, the P.R.R. values for palpitations, arrhythmias, atrial fibrillation, hypertension, and hypotension were lower or close to 1 for almost all sulfonylureas and glinides drugs.

Nateglinide had the highest P.R.R. values ≥2 for rhabdomyolysis, hypoglycaemia, palpitations, and arrhythmias vs. all other drugs, and the P.R.R. of muscular weakness was very close to the limit of significance.

Due to the lower number of A.D.R. of mitiglinide and tolbutamide, their data were not included in the P.R.R. calculations and other analyses. The P.R.R. were also calculated in the absence of ZOL data, and the risk profile did not change within sulfonylureas and glinides.

We therefore investigated the possible association of specific A.D.R. risk with KATP channel blocking actions. Linear correlation analysis showed that the IC_50_ of the drugs to block the KIR6.2-SUR1 is significantly correlated with the P.R.R. of hypoglycaemia and not with the IC_50_ to block the KIR6.2-SUR2A or KIR6.1-SUR2B channels. We also tested for linear correlation between the P.R.R. values of the KATP channel blockers and their IC_50_ to block the different recombinant KATP channels, but we failed to show a significant correlation between the P.R.R. of the osteonecrosis, myalgia, and KATP channel blocking action, all showing R < 0.5.

Multiple correlation analyses showed that muscular weakness and rhabdomyolysis are very well correlated with the IC_50_ of the KATP channel blockers to block the muscle KIR6.2-SUR2A channel. A significant correlation was also calculated between the combined atrial fibrillation and arrhythmias and the IC_50_ to block the muscle KIR6.2-SUR2A channel, and an even higher correlation was found with the IC_50_ to block the vascular KIR6.1-SUR2B channel by these drugs. Atrial fibrillation and hypertension appear to also be associated with the KIR6.1-SUR2B blocking actions (Table 3).

## 4. Discussion

We found that ZOL inhibits the KATP currents of soleus muscle fibers and primary bone culture at sub-nanomolar concentrations that highly express the KIR6.1 subunit and SUR2 [27,28], and it is a potent blocker of the recombinant KIR6.1-SUR2B channel, suggesting that the KIR6.1-SUR2B subunits are the drug targets in these tissues. We observed that *Kcnj11* expression in SOL was lower than in EDL, suggesting a reduced expression ratio *Kcnj11/Kcnj8* in favor of the functional presence of the KIR6.1 subunit in this muscle phenotype and thereby explaining the lower IC_50_ in SOL vs. EDL. Even so, the KATP currents of the EDL muscle fibers expressing high levels of KIR6.2 and SUR2A as well as those of the recombinant KIR6.2-SUR2A channel subunits were also significantly inhibited by ZOL. The finding that the P.R.R. of atrial fibrillation and arrhythmias as well as the muscular weakness and rhabdomyolysis are improved using multiple correlation equations supports the notion that multiple mechanisms, including KIR6.2-SUR2A channel blocking action, participate in this A.D.R. The contribution of the KIR6.1-SUR2B channel by ZOL to the atrial fibrillation and arrhythmias/hypertension is also supported by our multiple correlation analysis.

Despite no correlation being found between the osteonecrosis of the jaw and KATP channel blockers, the contribution of the KIR6.1-SUR2B channel blocking action by ZOL to this A.D.R. cannot be excluded. This channel is functionally expressed in vascular tissues and osteoblast playing a role in vascular perfusion, while the role in osteoblastogenesis is presently under investigation in our labs.

As expected by the high affinity toward hydroxyapatite, BPs are primarily taken and adsorbed into bone, where they can exert their pharmacological effects binding to *h*FPPS with antiproliferative actions in tumor cells and osteoclasts, and the mineralization of osteoblasts through different mechanisms involving ion channels and tyrosine kinase [5,6,15]. BPs distribution in bone is not homogenous but occurs mostly in bone regions with high turnover. Therefore, in vivo direct interaction between ZOL and KATP channels in bone and surrounding soft tissues (muscle and vessels) can easily be speculated. Even if the bone is the main site of distribution, BPs can also go to soft noncalcified tissues such as the liver, lung, kidney, and spleen [49]. The noncalcified tissue/plasma concentration ratio is very low, between 0.05–0.7; for the kidney, which is involved in drugs excretion, it may reach 6. BPs concentration in noncalcified tissue rapidly decreases with time, moving from 63% after 5 min from the dose to 5% after 1 h. On the contrary, drug concentration in bone increases continuously, reaching its peak after 1 h from the dose. This rapid redistribution of the BPs toward bones suggests that the soft tissues are in contact with the drug for a very short time. Even so, BPs can directly interact with these other organs, and direct interaction with channels in these districts cannot be excluded [50].

Our docking simulations result in binding with negative values of the binding energies of ZOL to KIR6.1/2 and SUR’s subunits, with the highest affinity shown for the SUR2A/B-Glib, SUR2A/B-ATP, and KIR6.1/6.2-ATP binding regions. The structural similarities between ZOL and ATP allow for binding interactions at the ATP binding region of the KIR6.1/2 subunits or at the ADP binding region of the SUR subunits.

Notably, it is known that ZOL inhibits *hFPPS*, leading to the accumulation of isopentenyl diphosphate, thereby converted to ATP isopentenyl ester and other ATP derivatives. These inhibit KIR6.1/2-SUR protein complexes with high specificity at the corresponding ATP binding regions, with ATP isopentenyl ester being found to also inhibit ADP/ATP carrier [51]. This has important implications for mitochondrial function and ADP/ATP carrier mediated mitochondrial apoptosis triggering [52,53]. This could represent an additional ZOL-mediated indirect mechanism of blocking KATP channels.

Thus, it is expected that the channels hosting the KIR6.1/6.2 and the SUR2A/B subunits are more sensitive to zoledronic acid compared to channels hosting KIR6.2 and SUR1, therefore being the first selective musculoskeletal KATP channel blocker available.

The interaction of ZOL with multiple binding sites on the KIR6.1/2 and SUR2B/A and the contribution of other ligands released into the cytosol help to explain the lower IC_50_ values and the sleepy slopes of the concentration-response curves of this drug on the native and recombinant KATP channels. The interaction of ZOL with the KIR6 and SUR2 present in the inner mitochondria membrane may contribute to the cytotoxicity of this drug in the cardiovascular system, as mitochondrial KATP channels play a significant role in cardio protection.

## 5. Conclusions

In conclusion, we first showed that ZOL binds to KIR6.1/6.2 and SUR2A/B subunits at subnanomolar concentrations and blocks the KATP channel, reducing native currents in the fibers and bone cells as well as the current of the recombinant channels expressed in cell line. This interaction provides a molecular explanation for some of the most common A.D.R. associated with ZOL therapy and provides insights for a better understanding of the drug’s effects on musculoskeletal and cardiovascular systems. Openers of KATP channels like nicorandil or diazoxide can be proposed to counteract the musculoskeletal and cardiovascular A.D.R. of ZOL.

## Figures and Tables

**Figure 1 pharmaceutics-13-01350-f001:**
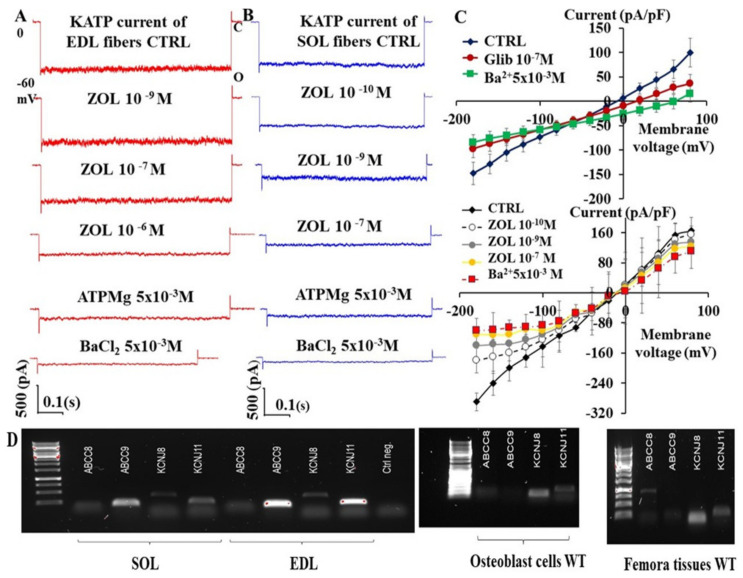
Effects of increasing concentrations of zoledronic acid (ZOL) on KATP currents recorded in extensor digitorum longus (EDL) (**A**) and soleus (SOL) (**B**) muscle fibers and on the current-voltage relationship of primary long bone cells in mice. (**A**,**B**) Sample traces from 1 patch per fiber of the KATP currents recorded in excised macro-patches from isolated fibers at −60 mV (Vm) in the presence of 150 mM high K^+^ ions concentrations on both sides of the membrane patches (O, open current level, C closed current level) in control condition (CTRL) in the absence of added nucleotide or modulators following the excision of the membrane patch from the fibers. The current and the drug action in excised patches was recorded at one voltage close to the resting potential due to the non-voltage-dependent nature of the KATP channel to avoid excessive membrane stress associated with patch isolation. The application of ZOL solution at 10^−9^ M concentration to the internal side of the membrane patch inhibited the current in SOL fibers by −83% and by −23% in EDL fibers in these patches. The KATP currents were further inhibited by ATPMg 5 × 10^−3^ M applied on the internal side of the macro patches in either fibers. The application of the BaCl_2_ solution, an unselective KIR blocker, at 5 × 10^−3^ M concentration fully inhibited the residual current, indicating that the ZOL inhibited KATP channel currents in these fibers. (**C**) Current/voltage relationship of primary long bone cells and effects of KATP channel blockers. The KATP currents were recorded in C-A patches from primary long bone cells in physiological asymmetrical K^+^ ions condition in the absence of CTRL, presence of glibenclamide (Glib), or increasing concentrations of ZOL solution, followed by BaCl_2_ solutions applied at the end of protocol period. Glib and ZOL markedly reduced the KATP currents at nanomolar concentrations in primary long bone cells, especially at negative membrane potentials. Each data point was obtained by 5–10 cells. (**D**) PCR experiments evaluating the expression of the KATP channel genes in Soleus (SOL) and Extensor digitorum longus (EDL) muscles and in femora bone and primary cells from calvaria mice. *Abcc9* and *Kcnj11* genes were identified in EDL muscle and in SOL muscle. The EDL and SOL muscles also showed the *Kcnj8* gene expression. A sample PCR gel shows an intense band at the *Kcnj8* gene, suggesting a high expression of KIR6.1 subunit and a low expression of the *Kcnj11*/KIR6.2 subunit in bone cells. SUR subunits were found. PCR experiments were performed in triplicate.

**Figure 2 pharmaceutics-13-01350-f002:**
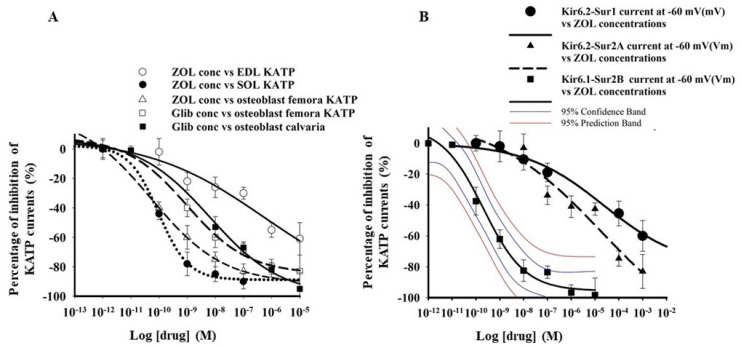
Concentration-response relationships of zoledronic acid (ZOL) against KATP currents of native tissues and the currents of recombinant subunits expressed in Hek293 cells. (**A**) A significant leftward shift of the curves was observed in Soleus (SOL) and primary long bone cells in the presence of an increasing concentration of ZOL on the log *x*-axis, indicating that the drug potently inhibited the channel subunits in these tissues. Glibenclamide (Glib) potently inhibited the KATP currents of primary long bone cells and calvaria bone cells. The percentage of inhibition of the currents caused by ZOL was calculated against the current recorded at −60 mV (Vm) in the presence of ATP 5 × 10^−3^ M in excised macro patches and BaCl_2_ in C-A patches −60 mV (ΔVm). The data point represents the mean ± E.S. of a minimum of 13 patches/cells/fibers. (**B**) A significant leftward shift of the concentration-response curves of the KIR6.1-SUR2B current subunits was observed in the presence of the increasing concentration of ZOL on the log axis. The percentage inhibition of the currents caused by ZOL was calculated against the current recorded in whole cells at −60 mV (Vm) in physiological conditions in the presence of BaCl_2_. The data point represents the mean ± E.S. of a minimum of 15 cells.

**Figure 3 pharmaceutics-13-01350-f003:**
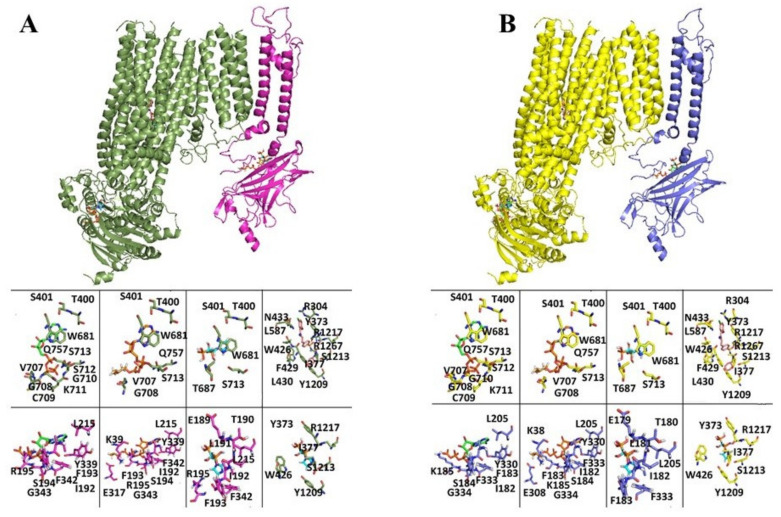
3D comparative models of KIR6.1-SUR2B human protein complex (**A**). Lateral views of SUR2B (dark green cartoon)-KIR6.1 (magenta cartoon) human protein complex is reported in cartoon representation. In the panels below, within the first row, ATP (green sticks), ATP-IPP (orange sticks), and zoledronic acid (cyan sticks) are reported in the SUR2B ATP binding region (dark green sticks), whereas Glib (pink sticks) is reported in the SUR2B—Glib binding region (in dark green sticks). In the panels below, within the second row, ATP (green sticks), ATP-IPP (orange sticks), and zoledronic acid (cyan sticks) are reported in the KIR6.1 ATP binding region (magenta sticks). In addition, Zoledronic acid is reported in the SUR2B-Glib binding region (dark green sticks). (**B**) KIR6.2-SUR2A 3D comparative models of KIR6.2-SUR2A human protein complex. Lateral views of SUR2A (yellow cartoon)-KIR6.2 (dark blue cartoon) human protein complex is reported in cartoon representation. In the panels below, within the first row, ATP (green sticks), ATP-IPP (orange sticks), and zoledronic acid (cyan sticks) are reported in the SUR2A ATP binding region (yellow sticks), whereas Glib (pink sticks) is reported in the SUR2—Glib binding region (in yellow sticks). In the panels below, within the second row, ATP (green sticks), ATP-IPP (orange sticks), and zoledronic acid (cyan sticks) are reported in the KIR6.2 ATP binding region (yellow sticks). In addition, Zoledronic acid is reported in the SUR2A-Glib binding region (yellow sticks).

**Figure 4 pharmaceutics-13-01350-f004:**
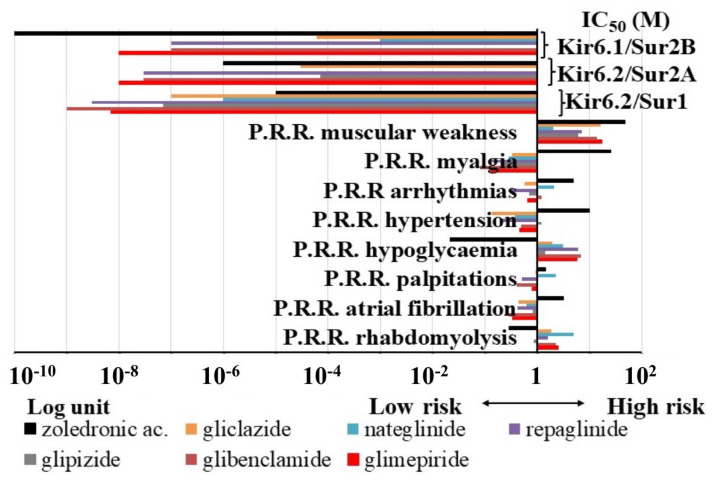
Distribution of the Proportional Reporting Ratio (P.R.R.) of different risks vs. the IC_50_ values of the drugs against the different recombinant KATP channel subunit combination expressed in cell lines. P.R.R. values lower than 1 indicate low risk; values higher than 1 indicate higher risk. Zoledronic acid shows a high risk for all A.D.R. apart from hypoglycaemia and rhabdomyolysis, and the highest potency against the vascular KIR6.1-SUR2B channels vs. other drugs. The sulfonylureas and glinides show a higher risk for hypoglycemia and muscular weakness, and nateglinide shows additional risks for cardiac A.D.R. and rhabdomyolysis. The sulfonylureas and glinides show the highest potency against the pancreatic KIR6.2-SUR1 channels vs. zoledronic acid except for nateglinide, which is the less potent drug within the glinides and sulfonylureas.

**Table 1 pharmaceutics-13-01350-t001:** Fitting parameters of the concentration-response relationship of zoledronic acid against native skeletal muscles fibers and primary bone cells and recombinant KATP channel subunits expressed in Hek293 cells line.

Drugs Cell Types Patch Clamp Configuration	Imax %	IC_50_ (M)	Slope
Zoledronic acid EDL fibers excised macropatches	−100.1 ± 9.3 *n* = 17 fibers	1.2 ± 1.4 ×10^−6^	0.38 ± 0.07
Zoledronic acid SOL fibers excised macropatches	−99.4 ± 5 *n* = 21 fibers	* 2.1 ± 3.7 × 10^−10^	0.29 ± 0.05
Zoledronic acid Femora primary bone cells Cell-attached patches	−104.8 ± 8.7 *n* = 21 cells	* 1.6 ± 2.8 × 10^−10^	0.31 ± 0.07
Glibenclamide Femora primary bone cells cell-attached patches	−88.32 ± 21 *n* = 13 cells	* 5.07 ± 0.5 × 10^−9^	0.4 ± 0.08
Glibenclamide Calvaria primary bone cells cell-attached patches	−93 ± 14 *n* = 18 cells	* 1.02 ± 0.6 × 10^−8^	0.5 ± 0.03
Zoledronic acid Hek293 cells- KIR6.2-SUR2A Whole-cell	−83.7 ± 1.2 *n* = 15 cells	7.1 ± 3.1 × 10^−6^	* 0.12 ± 0.7
Zoledronic acid Hek293 cells -KIR6.1-SUR2B Whole-cells	−94 ± 4.8 *n* = 15 cells	* 3.9 ± 2.7 × 10^−10^	0.59 ± 0.04
Zoledronic acid Hek293 cells -KIR6.1-SUR1 Whole-cells	* −40 ± 8.8 *n* = 3 cells	/	/
Zoledronic acid Hek293 cells -KIR6.2-SUR1 Whole-cells	* −60.13 ± 4.1 *n* = 15 cells	2.1 ± 4.1 × 10^−5^	* 0.18 ± 0.09

EDL, Extensor digitorum longus; SOL, Soleus. Whole-cell and cell-attached KATP currents were recorded in physiological condition, KATP currents in native fibers were recorded in excised macro-patches in the presence of symmetrical high K^+^ ions on both sides of the membrane patches. * Data significant difference vs. other groups by One WAY ANOVA with variance-ratio F > 1.621 for *p* < 0.05.

**Table 2 pharmaceutics-13-01350-t002:** Affinities of the reported ligands for the indicated binding regions within the investigated KIR-SUR protein complexes.

		Ligand Binding Affinity (G (Kcal/mol))		
		KIR ATP Binding Region	SUR ATP Binding Region	SUR Glib Binding Region
**HsKIR6.1SUR2B**	ATP	−5.9	−6.3	ND
	Zoledronic acid lowest rmsd pose	−6.3	−5.3	−6.5
	Zoledronic lowest binding energy pose	−6.6	−5.3	−6.4
	ATP_isopentenyl ester	−6.6	V6.3	ND
	Glib	ND	ND	−9.1
**HsKIR6.1SUR1**	ATP	−5.9	−6.3	ND
	Zoledronic acid lowest rmsd pose	−4.3	−5.3	−4.4
	Zoledronic lowest binding energy pose	−4.3	−5.3	−4.4
	ATP_isopentenyl ester	−6.6	−6.4	ND
	Glib	ND	ND	−9.1
**HsKIR6.2SUR2A**	ATP	−6.1	−6.3	ND
	Zoledronic acid lowest RMSD pose	−5.4	−5.3	−6.1
	Zoledronic lowest binding energy pose	−5.4	−5.6	−5.9
	ATP_isopentenyl ester	−6.2	−6.3	ND
	Glib	ND	ND	−9.2
**HsKIR6.2SUR1**	ATP	−6	−6.3	ND
	Zoledronic acid lowest RMSD pose	−5.4	−5.3	−4.2
	Zoledronic lowest binding energy pose	−4.8	−5.4	−4.4
	ATP_isopentenyl ester	−6.2	−6.4	ND
	Glib	ND	ND	−9.9

**Table 3 pharmaceutics-13-01350-t003:** Correlation analysis of IC_50_ of the KATP channel blockers to inhibit the currents of recombinant channels in cell lines vs. their Proportional Reporting Ratio (P.R.R.).

P.R.R. of KATP Channel Blockers	IC_50_ (M) of KATP Channel Blockers	Equations and Coefficient of Correlation
hypoglycaemia	KIR6.2-SUR1	y = aln(x) + b a = −0.708 b = −8.2542 R^2^ = 0.7849
muscular weakness rhabdomyolysis	KIR6.2-SUR2A	y = b0 + blnx1 + b2x^2^ b1 = −9.114 × 10^−6^ b2 = 3.505 × 10^−5^ b0 = −3.493 × 10^−5^ R^2^ = 0.83249
atrial fibrillation arrhythmias	KIR6.2-SUR2A KIR6.1-SUR2B	b1 = 4.0144 × 10^−5^ b2 = −2.708 × 10^−5^ b0 = 2.213 × 10^−5^ R^2^ = 0.602 b1 = −6.878 × 10^−4^ b2 = 4.342 × 10^−4^ b0 = 9.518 × 10^−5^ R^2^ = 0.699
atrial fibrillation hypertension	KIR6.1−SUR2B	b1 = 1.6465 × 10^−3^ b2 = −4.862 × 10^−4^ b0 = −3.851 × 10^−4^ R2 = 0.6177

## Data Availability

The data supporting the reported results are available for further evaluation.

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
