# Peer review of "Zoledronic Acid as a Novel Dual Blocker of KIR6.1/2-SUR2 Subunits of ATP-Sensitive K+ Channels: Role in the Adverse Drug Reactions"

_pharmaceutics, 2021, doi:10.3390/pharmaceutics13091350_

Round 1

Reviewer 1 Report

The manuscript by Fatima et al. investigates the role of ATP-sensitive potassium channels (KATP) in adverse drug reactions caused by zoledronic acid (ZOL).  The authors determined ZOL was able to inhibit KATP channel currents in fast and slow twitch muscle fibers, which express the Kir6.2/SUR2 subtypes of these channels (Kcnj11 and Abcc9, respectively).  ZOL was a more potent inhibitor of KATP channels in the slow-twitch fibers vs the fast-twitch fibers. The authors also tested the inhibitory potential of ZOL on primary bone cells compared to a known KATP channel antagonist, gliblenclamide (Glib).  Primary bone cells appear to express the Kir6.1/SUR1 subtypes of KATP channels, and appear to be as sensitive to ZOL as the slow twitch (Soleus, SOL) muscle fibers.  The authors also performed docking studies in silico to determine any possible binding sites of ZOL on KATP channels.  By aligning the protein sequences of human SUR1, SUR2, SUR2B, KIR6.2, and KIR6.1 with the crystal structures of Kir6.2 from M. musculus and Sur1 from Mesocricetus auratus to create a 3D structural model. From here, the authors determined potential docking sites for ZOL in the ATP, ADP and Glib binding sites within the SUR protein. Finally, the authors performed a linear correlation analysis to determine if reported adverse drug reactions of ZOL and other KATP channel inhibitors are linked to the IC50 values of different Kir6.1/2-Sur1/2A/2B subtypes. The authors ultimately found IC50 values were strongly correlated with the proportional reporting ratio (PRR) of a given adverse event.

The data provide a molecular rationale to correlate adverse drug reactions on muscular and cardiovascular systems with KATP channel inhibition.  With several revisions, the manuscript should be ready for publication in Pharmaceutics.

Comments:

  1. There is no section within the Materials and Methods for the PCR performed within the manuscript.
  2. Protein names should conform to organism-specific formatting guidelines. In general, for rodents, Gene symbols are italicized, with only the first letter in upper-case (e.g., Gfap). Protein symbols are not italicized, and all letters are in upper-case (e.g., GFAP).   This is different than proteins for humans.  Could the authors please check:  Page 4 line 154 and page 6 lines 255, and 274-289.  The figure legends for Figure 1 and S1 should also be checked.
  3. On page 4 line 197, root-mean-square-deviation needs the acronym added as RMSD is used later in the manuscript.
  4. For the Pharmacovigilance section, is all the data from the EudraVigilance site? Otherwise citations are needed for all the data stated in the 3.1.2. section.
  5. The primer sequences and the product lengths are not provided for Figure 1.
  6. The legend for Figure 1 states, “Effects of increasing concentrations of ZOL on KATP currents in skeletal muscle and on primary long bone cells…” Are the long bone cell currents panel 1C? 
  7. Was the percent identity information within section 3.1.1. Docking analysis taken from the NCBI Blast webpage or is the percent identity information gained from the protein alignments in ClustalW? If it’s from NCBI, this should be cited.
  8. Figure 4 would be easier to read if each IC50 for each KATP channel subtype were separated. A regression analysis (e.g. IC50 vs PRR) might be more clear than a bar graph?  The x-axis does not have units or a scalebar.
  9. Can more context be given to Figure S1C? Is this an entire 10cm plate stained with alizarin stain? Or, just a single well of a multi-well plate?

Author Response

1) that's right we introduced the PCR related paragraph

2) all gene and protein nomenclature was revised according to guideline

3) ok

4) all data were extracted from EUdraVigilance

5) The primer sequences of PCR gene was reported in the revised methods

6) The legend of  figure 1 was revised accordingly 

7) ClustalW was used for calculating the % of identical residues between the compared protein sequences and the dedicated reference was indicated

8) The reviewer is absolutely right we lost the x-axis in the previous figure and the data were not clearly presented. We replaced it with a new one separating the IC50 values and PRR and with the x-axis, we thank the reviewer for the suggestion and the new bar graph shows the IC50 values grouped and separated from the PRR values. The bar graph distribution better evidence the risk of ADR associated with ZOL  when PRR >2 and the lower risk in case of PRR <2, however, we should note that we do have several correlation analysis plots, in case of linear regression, and table 3 reports the relative equation correlating indeed the PPR against the IC50 of different drugs and the calculated parameters. We should also stress that we tested several possible combinations of PRR and IC50 and were negative, we only reported those showing R >0.5, for instance, we failed to find a significant correlation between the IC50 of the drug in blocking KATP channels and the osteonecrosis of the Jaw.

9) No that is the full 10 cm  plate, we normally do it on this large plate during bone cells isolation to monitor osteoblasts formation during time and PCR of gene on cell pellets, if either test are positive the cells are used for patch 

Reviewer 2 Report

The manuscript ID pharmaceutics-1320039 entitled “Zoledronic acid as a novel dual blocker of Kir6.1/2-Sur2 subunits of ATP-sensitive K+channels: role in the adverse drug reactions “by Fatima et al is quite interesting and suitable to the pharmaceutics journal. The authors investigated the action of ZOL by performing in vitro patch-clamp experiments on native KATP channels in murine skeletal muscle fibers, bone cells, and recombinant KATP channels expressed in cell lines, and by in silico 23 docking. In the results, authors were observed that ZOL fully inhibited the KATP currents recorded in excised macro-patches from Extensor 24 digit forum longus (EDL) and Soleus (SOL) muscle fibers with an IC50 of 1.2±1.4 x10-6and 2.1±3.7x10-25 10 M, respectively, and the KATP currents recorded in cell-attached patches from primary long 26 bone cells with an IC50 of 1.6±2.8 x10-10 M. ZOL fully inhibited whole-cell KATP channel current of recombinant Kir6.1-Sur2B and Kir6.2-Sur2A channel subunits expressed in HEK293 cells with an IC50 of 3.9±2.7x10-10 M and 7.1±3.1x10-6 M, respectively. The rank order of potency in inhibiting the 29 KATP currents was: Kir6.1-Sur2B/ SOL-KATP/ osteoblast-KATP > Kir6.2-Sur2A/EDL-KATP>>> 30 Kir6.2-Sur1 and Kir6.1-Sur1. Further, the authors performed docking and the results revealed that the drug binds to the ADP/ATP sites on Kir6.1/2 and Sur2A/B and on the sulfonylureas site. Finally, the authors concluded that the IC50 of ZOL inhibits the Kir6.1/2-Sur2A/B channels is correlated with its musculoskeletal and cardiovascular risks. However, there are some important concerns to be addressed before publication as follows, 

1)    In the abstract there is too much background information, authors may need to brief up and include the docking results 
2)    Page 2: line 72-74: the correlation is missing the adverse effect of Zoledronic acid and the physiopathological role of KATP------So, ZOL only control musculoskeletal based on the target KATP? Or ZOL also causes cardiovascular risks
Among these, gastric irritation, osteonecrosis of the jaw, atypical femoral fractures, esophageal cancer, atrial fibrillation, severe musculoskeletal pain and weakness, and atypical femur fractures are observed in the disease patients population [13] ……….and KATP channels have a well-known physiopathological role in regulating vascular tone, cardiac excitability, and neuronal firing, the insulin release from the pancreas, muscle fatigue and pain, neurodegeneration [16-23], is also involved in not familial diseases and in rare KATP chan-96 nelopathies [19, 24].
3)    In references 6, 27-32, 37, 38 authors stated that all are previously described….if so, what is the alteration in the experiments, explain in briefly or else it is just for increasing self-citations
4)    Reference missing – Modeller, Autodock4.2.
5)    Write the sentence clearly and include citations properly---------- “All the generated 3D all-atom models were energetically minimized by using the Rosetta “relax” application within the Rosetta “scoring and prep” tools [39][40][41]. The obtained final models were examined in VMD, PyMOL, and SPDBV by 183 visual inspections searching for putative unsolved clashes [37] [38].”
6)    How about the structural quality of the models? Ramachandran plot? ERRAT?
7)    What is S.O.C.? and what are the putative unsolved clashes?
8)    Excel soft-206 ware (Microsoft 10.00).------I think excel is a Microsoft package
9)    Whether the data obtained from triplicates or duplicates of experiments? Mention it on statistical analysis
10)    Conclusion might be improved which highlights the results of the study and include future perspectives
11)    Many typos, improper usage of abbreviations, and inconsistent reference formats---needs to be improved
12)    The language of the manuscript should be in journal standard

In my overall observations, this is a good interesting study but the authors should revise the manuscript according to the comments.  

Author Response

1) the abstract was revised including docking data and deleting backgrounds 

2) we revise the paragraph trying to better correlate the ZOL effects and KATP channel function

3) As suggested by the reviewer the references 31, 32,  were deleted to eliminate unnecessary citations. References 37 and 38 were indicated as case studies. Indeed, the employed procedure was described in a more detailed way in the manuscript corresponding to reference [38]. However, the complete comparative procedure was developed in a stand-alone package, the object of a patent request, as the reviewer can read in the acknowledgment section of the paper [38].

4) Modeller and Autodock4.2 references were added accordingly to the reviewer indications

5)  We thank the reviewer for pointing it out. The references about the indicated software have been added.

6) We thank the reviewer for this question that helped us to make clearer the 3D comparative modeling section. It is known that “Modeller” implements the modeling by satisfaction of spatial restraints. The procedure is conceptually similar to the protocol used for the determination of protein structure by using NMR based approaches. The restraints are defined by assuming that the distance between aligned residues in the protein template and in the protein to be modeled are similar. These restraints are known as homology-derived restraints and are usually supplemented by stereochemical restraints on bond lengths, bond angles, and so on, which are obtained by relaxing the 3D model in a molecular mechanics force field.  This info are provided in the newly provided “MODELLER” reference.

Furthermore, we indicated that all the generated 3D all-atom models were relaxed by using the indicated Rosetta tools. The structural properties of the models with the best energy function were checked by using the Whatif webserver package (i.e., to check they, f and Cb angles of the residues). A dedicated sentence with a reference to Whatif Webserver was added.

7) SOC is System Organ Class and it is now defined. We thank the reviewer for this question. About the “putative unsolved clashes”, it may happen that a 3D model can result globally energetically minimized, but some side chains may still form a so-called “clash”. I.e., it may happen that the side-chain of lysine may cross the aromatic ring moiety of the side chain of a phenyl-alanine. In some rare cases, some programs can fail to reveal clashes and a visual/manual inspection may help in further reducing those rare cases. Clashes are defined in the cited review: https://www.sciencedirect.com/science/article/pii/S1570963910001068?via%3Dihub that now is recalled also in the revised version of the manuscript in paragraph 2.4 where clashes are cited (see lines 180-190).

8) Yes it is and is now defined.

9) Data were in triplicate.

10) The conclusion has been revised with some perspectives evidenced; 11) The reference list was revised, as well as the manuscript

12) the language and editing has been also adapted to the Journal  

Reviewer 3 Report

Review according manuscript: “Zoledronic acid as a novel dual blocker of Kir6.1/2-Sur2 subunits of ATP-sensitive K+channels: role in the adverse drug reactions” by Fatima Maqoud, Rosa Scala, Vincenzo Tragni, Ciro Leonardo Pierri, Maria Grazia Perrone, Antonio Scilimati, Tricarico Domenico.

Manuscript ID: pharmaceutics-1320039

In these study, authors have been tested the effects of ZOL on KATP channels by performing patch-clamp experiments in native murine fast-twitching and slow-twitching muscle fibers, primary bone cells culture, and recombinant channel subunits expressed in heterologous cells system. Furthermore, they performed docking studies to evaluate the in silico interaction of ZOL with KATP channel subunits and to identify the possible binding site. Finally, a pharmacovigilance investigation was performed to evaluate the musculoskeletal A.D.R. of ZOL associated with KATP channel interaction.

The topic is interesting and important, and the findings have a high impact to broad ranged readers. The experiments are carefully designed and well performed. I believe the scientific merit is worthy for publication in a prestigious journal such as Pharmaceutics. Generally speaking, the paper is sound; however there are some problems.

Abstract:

The novelty of the study is not clear. Please rewrite or add one or two sentence to the abstract.

Main body of the manuscript

Introduction:

Line 85-88. Sentence according preliminary, unpublish data should be removed from Introduction section and eventually used in Results section.

I suggest adding a paragraph regarding the structure of the KATP channel (subunit composition Kir/SUR). In my opinion it is important especially in the context of molecular modeling analysis presented in the manuscript.

Material and methods:

Please provide catalog no for KATP channels modulators and especially for Ba2+ ions.

Composition of the patch-clamp solution is not clear. Ionic compositions are the basis for the analysis of electrophysiological data.

Results:

The protocol of the experiments presented in fig 1. is not clear. Additionally, why authors presented only one “current” at -60 mV (fig 1A, 1B)?

From how many repetitions of experiments under the same conditions the curves were prepared (fig 1C)? The same PCR experiments.

The figures are not in the place where they are described. This makes reading the manuscript difficult.

Discussion:

Line 454. What does mean “…”?

Reading the discussion, you can hardly see what constitutes the novelty of the research undertaken?

Line 475. “Thus, it is expected that the channels hosting the Kir6.1/6.2 and/or the Sur2A/B sub-475 units should be more sensitive to zoledronic acid, compared to channels hosting Kir6.2 476 and Sur1”. What is the conclusion?

Can the observed dependencies translate into the regulation of the KATP channel present in the mitochondria? If so, this should be discussed.

Author Response

We thanks the reviewer for her/his positive comment's on our manuscript

1) the abstract has been revised as requested

2) the sentence in the introduction has been deleted from the introduction and moved in the results, we also add a molecular description of the KATP channel in the introduction as suggested 

3) the cat number of the drugs were indicated as well as the composition of the electrophysiological solution that was not reported  and we are sorry for this

4) the legend of figure 1 was revised explaining also the protocol applied, in the excised patch we normally test the drug effect at one voltage close to the resting state due to the not voltage-dependent nature of the channels, this avoids membrane stress and seal instability associated with multiple stimuli that is instead much well controlled in whole-cell and cell-attached patches. This is now reported in the legend.

Despite the gel are referred to as a single run, the PCR experiments are normally performed in triplicate and this is now clarified in the legends and methods section 

This is right we moved all figures and tables in the right place

The discussion was revised trying to better stress the novelty of our finding and the possible involvement of mitochondrial in the generation of the ZOL dependent cardiovascular ADR 

Round 2

Reviewer 1 Report

The edits from the authors have much improved the readability of this manuscript.